# Ultrasonographic and pathological correlation of asymmetric retroareolar density on mammogram

**Anamika Jha[1], Ranjit Kumar Chaudhary[2]\*, Shreya Shrivastav[3], Umesh Khanal[1]**

1 Department of Radiodiagnosis and Imaging, Tribhuvan University Institute of Medicine, Maharajgunj, Kathmandu, Nepal, 2 Department of Radiology, St Vincent's Medical Center, Bridgeport, CT, United States of America, 3 Department of Pathology, Tribhuvan University Institute of Medicine, Maharajgunj, Kathmandu, Nepal

\* ranjit.chaudhary@hhchealth.org

## Abstract

**Data Availability Statement:** All relevant data are within the paper and its Supporting Information files.

### Background

Retroareolar region refers to the region within two centimeters from the nipple and/or involves the nipple-areolar complex on mammogram. In this study, we graded asymmetric retroareolar density on mammography and determined the underlying cause.

### Objectives

To identify and grade retroareolar densities and evaluate characteristics of lesion using ultrasonography and histopathology.

### Methods

Mammograms with asymmetric retroareolar density done in our tertiary care hospital were included. Retroareolar density was categorized into three grades based on morphological appearance in mammography. Sonography was performed in all patients and tissue diagnosis was obtained for suspicious lesions.

### Results

Of the 100 patients included in the study, most of the patients with mammographic grade 1, grade 2 and 3 retroareolar asymmetry had normal sonography, pathologically proven mastitis and invasive ductal carcinoma, respectively. Presenting indication usually was diagnostic (n = 87), lump being most common. Benign (58%) diagnosis was more often present, with equal number of normal studies and malignancies (21%). Frequently pathologically proven malignant lesions (n = 17) had grade 3 asymmetry and none were grade 1. Invasive ductal carcinoma was the most common malignancy while mastitis the most common benign disease.

**Funding:** The authors received no specific funding for this work.

**Competing interests:** The authors have declared that no competing interests exist.

## Conclusions

Grade I retroareolar asymmetric density on mammography was normal or had a benign etiology while grade 2 or 3 asymmetric density had underlying pathology, often malignancy.

## Contribution

Grading retroareolar density in mammogram may improve the evaluation of retroareolar region and increase emphasis on higher grades.

## Introduction

Retroareolar region is the region within two centimeters from the nipple and/or involves the nipple-areolar complex on mammogram [1]. This area includes terminal ends of lactiferous ducts and sinuses opening into the nipple, Montgomery glands and Morgagni tubercles [2, 3]. It may be affected by multiple benign or malignant pathologies and 8% of all breast cancers occur in this region [4].

Though clinically palpable, the cancers in this region may be missed on imaging [5]. Sonographic evaluation of this region is difficult due to the variable acoustic shadows caused by the air trapped in the areolar crevices [6, 7]. Asymmetry seen in retroareolar region on mammogram may be under or over diagnosed. Underlying cause needs to be evaluated with further investigations to ease patient anxiety or prompt management as necessary. In this study, we graded mammographic asymmetric retroareolar density and determined the underlying cause.

## Materials and methods

A hundred patients who underwent mammography at our center over a period of two years from January 2019 to June 2021, meeting inclusion criteria were included in this study. Ethical approval was obtained from the Institutional Review Board [42(6-11-E)$^2$/5/5/076]. The mammograms included two basic projections (cranio-caudal & medio-lateral oblique) of the breasts obtained with the digital mammographic unit in the department. Images were evaluated visually on workstation by two radiologists with more than 10 and 4 years of experience. Overall breast density was evaluated according to the American College of Radiology Breast Imaging and Data System (ACR BIRADS) atlas [7]. Retroareolar asymmetric density was described as that present only on one side on mammogram in retroareolar region, that is, within two centimeter of the nipple, though, which may not be limited to this area. According to the ACR BIRADS definition, asymmetry was defined as that seen only on one view and focal asymmetry when seen in both the views [7, 8]. Further, it was graded as follows:

Grade 1: Asymmetry (Fig 1)
Grade 2: Focal asymmetry (Fig 2)
Grade 3: Mass (Figs 3 and 4)

Patients with defined asymmetric retroareolar density willing to undergo sonography and participate in the study were included. Written informed consent was obtained and confidentiality maintained. Those with diffuse or developing asymmetry, lost for follow up with sonography or unavailable final pathological diagnosis for suspicious masses were excluded. Sonography was done in all patients by same radiologist. Both mammography and sonography were categorized into BIRADS categories. Tissue diagnosis was obtained when the lesion was suspicious, that is, BIRADS 4 or 5, by various techniques including fine needle aspiration

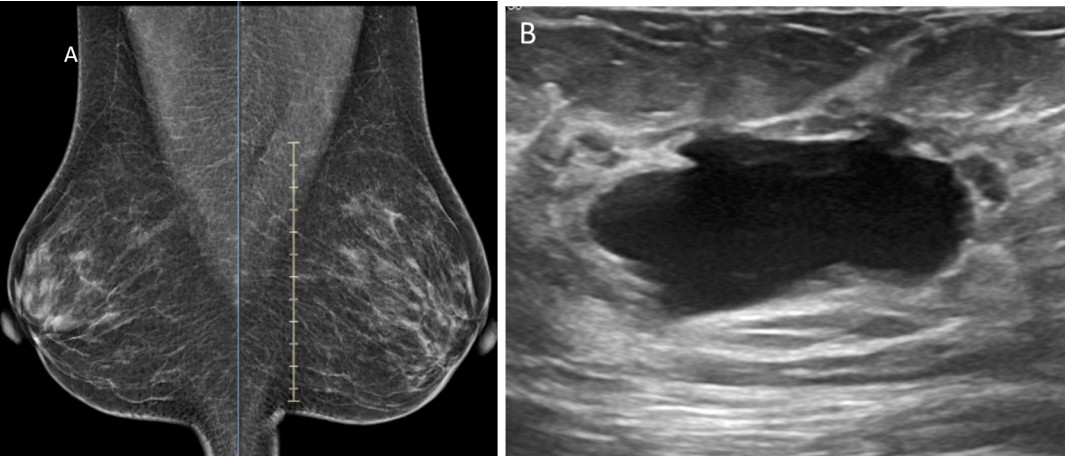

**Fig 1.** A female in her 50s who came for screening mammogram A) MLO view mammogram shows grade 1 asymmetric density in right retroareolar region B) Greyscale US image of corresponding region shows cystic lesion.

cytology, core needle biopsy and excisional biopsy. The findings were documented in prede-signed datasheet and analyzed using excel and Fisher's Exact test was performed using SPSSv20 software.

## Results

A hundred patients with age ranging from 29 to 76 years and mean age of 48 were included in our study with maximum number of patients (41%) in the 40–50 years group. Most of the

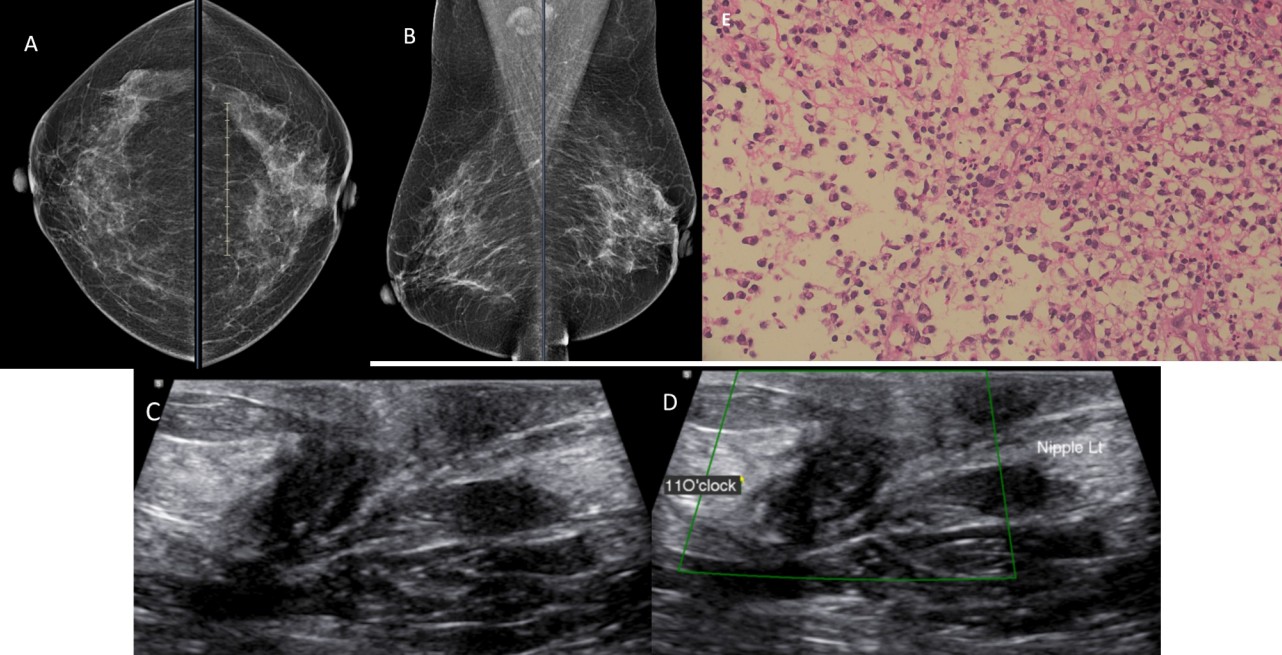

**Fig 2. A female in her 50s underwent screening mammogram.** A & B) CC and MLO views of breast show grade 2 focal asymmetry in left subareolar region. C & D) Greyscale and Color Doppler ultrasound images of left breast in retroareolar region shows ill-defined hypoechoic lesion without significant vascularity. E) Histopathological evaluation of biopsy specimen showed diffuse dense lymphoplasmacytic inflammatory infiltrates along with macrophage suggestive of subareolar mastitis. HEx200.

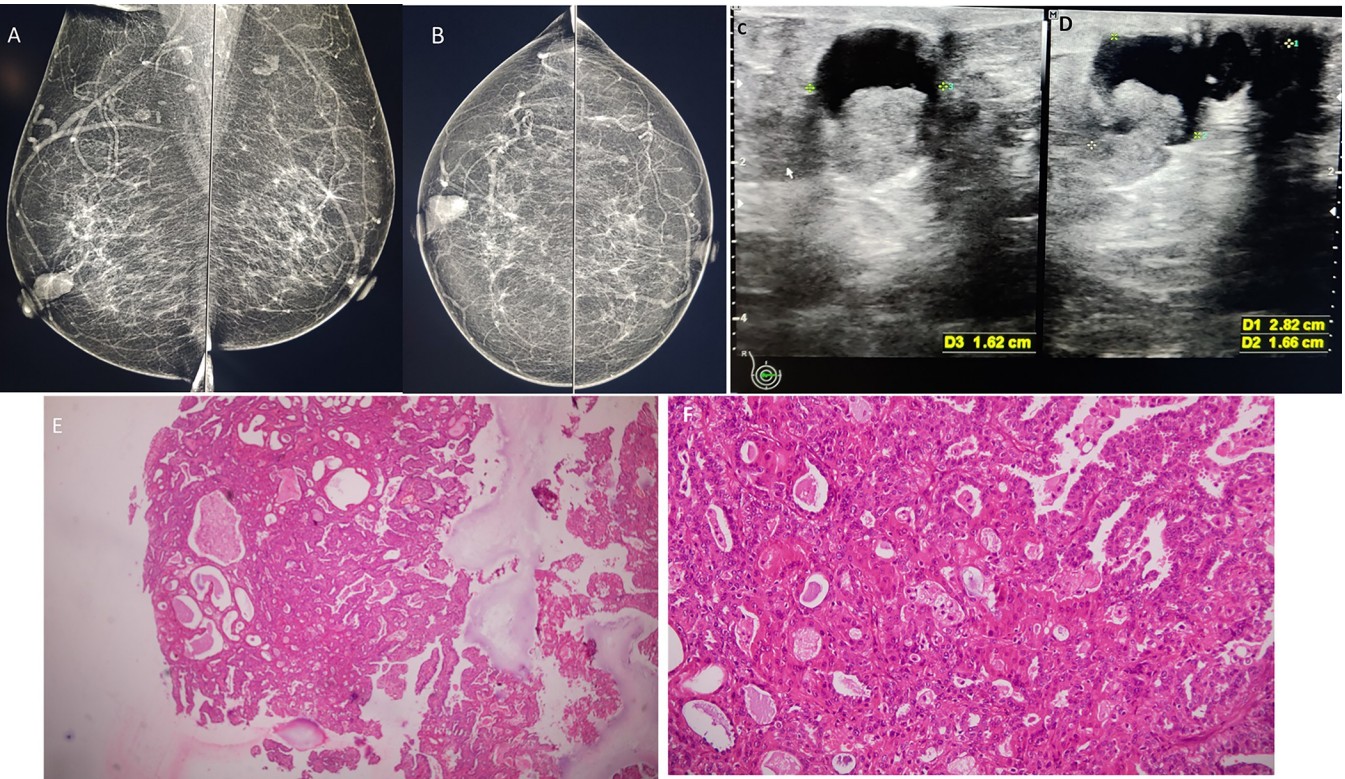

**Fig 3.** A female in her late 50s with pathological right nipple discharge A & B) MLO and CC views of breast show grade 3 retroareolar density with mass like appearance. C & D) Sonography shows a complex cystic mass with solid component E) Biopsy showed a polypoid lesion with branching papillae and dilated ducts. HE x40 F) Back-to-back ducts, are seen all of them showing myoepithelial cells. Foamy macrophages are also seen within the fibrovascular cores. Final diagnosis: Papilloma. HE x200.

patients were symptomatic (87%) while 13% of the patients were asymptomatic with screening mammograms. Lump was the most common symptom seen in 64 patients followed by mastalgia in 16 patients (Table 1).

The density distribution was as follows: 51% had heterogeneously dense parenchyma, 24% scattered fibroglandular densities, 14% dense and 8% fatty. Asymmetric density was seen more often on the right side (59%) than left and in patients with breast density b and c. Grade 1 asymmetry was seen in 33% of the patients, 39% had grade 2 and 28% had grade 3 asymmetry. Breast density distribution in the different grades of asymmetry was as shown in the Table 2.

Most of the patients with mammographic grade 1, grade 2 and 3 asymmetry had normal sonography 16 (76.19%), pathologically proven mastitis 31 (53.44%) and invasive ductal carcinoma 17 (80.95%), respectively. Fifty-eight percent of asymmetric retroareolar density were benign, with equal number of normal and malignant ones 21% each (Fig 5). Of the 21 pathologically proven malignant lesions, most (80.95%) had grade 3 asymmetry and none had grade 1 asymmetry. Most of the benign lesions (31 of 58) showed grade 2 asymmetry while 16 of 21 normal studies had grade 1 asymmetry on mammogram. Invasive ductal carcinoma was the most common malignancy seen in 19 of 21 patients while mastitis with or without abscess (18 of 58) followed by cystic disease (17 of 58) were the most common benign lesions (Table 3). Two patients presented with retracted nipple had grade 1 asymmetry on mammogram with a final normal diagnosis in one and benign cystic disease in the other. Using Fisher's exact test, the odds of grade 1 asymmetry to be a malignant lesion was significantly low (p<0.001).

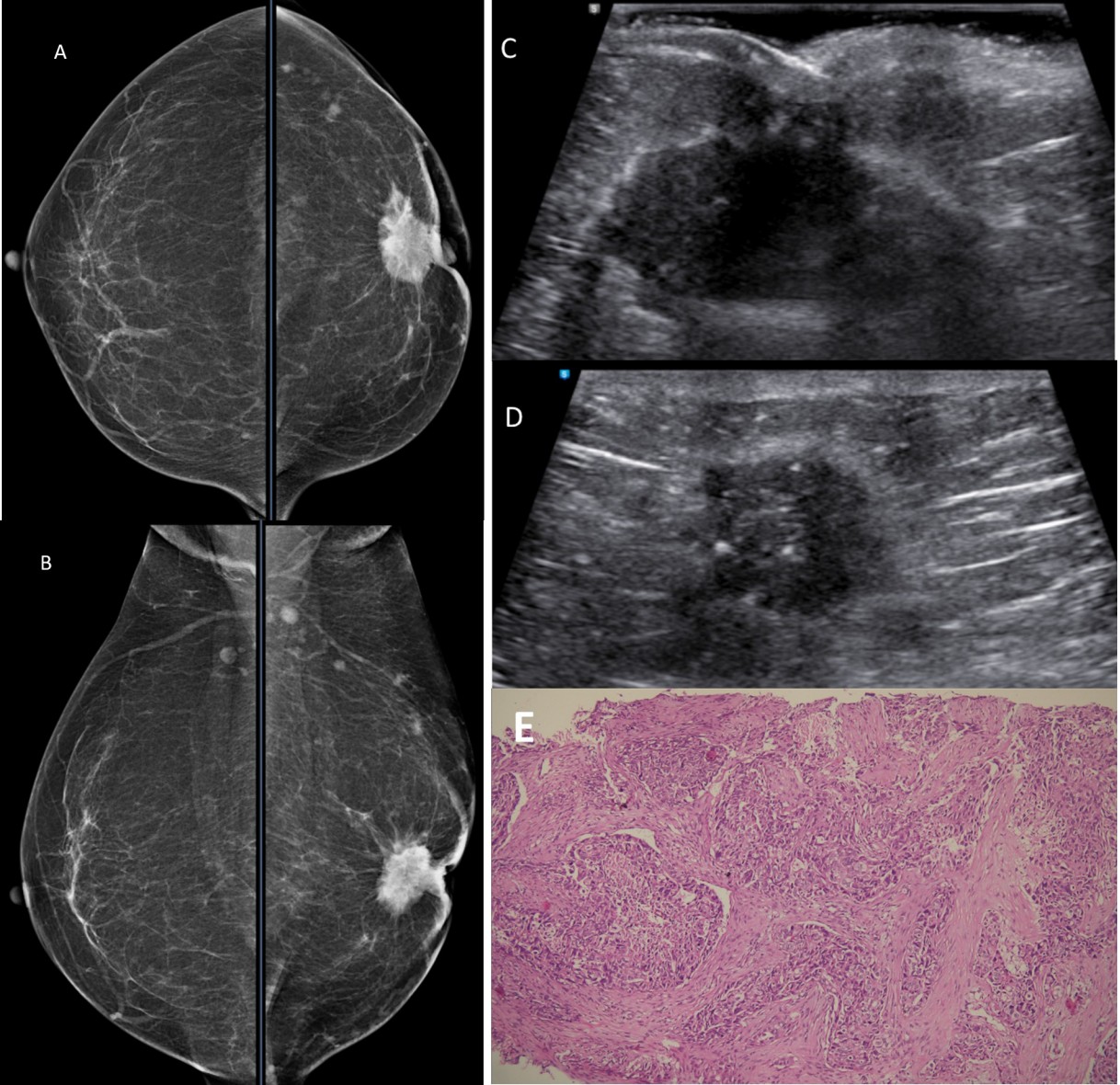

**Fig 4.** A female in her 40s with left subareolar mass A & B) CC and MLO views of breast show grade 3 retroareolar density in the form of spiculated mass with nipple retraction and adjacent skin thickening. C & D) US shows Irregular hypoechoic subareolar mass with microlobulated margins and nipple retraction. E) Biopsy shows large lobules of tumor cells with solid pattern and few tubules. Small nests, cords and comedo necrosis is noted. Final Diagnosis: Invasive Ductal Carcinoma. HE x 200.

The mean age of patients who had malignancy on histopathology was 58 years. All the patients with final diagnosis of malignant lesion presented with breast lump including four patients who had grade 2 retroareolar asymmetry. Eleven patients with grade 3 and one patient with grade 2 retroareolar density with final diagnosis of malignancy had suspicious calcification on mammogram. Of the 21 patients all except one were diagnosed based on biopsy results while one patient had excisional biopsy. Of the patients with malignancy, six underwent mastectomies while the rest of the patients had breast conservative surgery. Of these six patients who had mastectomies, two were grade 2 asymmetry.

**Table 1. Indications of mammogram in the study.**

| Indication | Percentage |
|---|---|
| Lump | 64 |
| Lumpiness | 1 |
| Mastalgia | 16 |
| Abscess | 1 |
| Areolar sinus | 1 |
| Nipple discharge | 2 |
| Retracted nipple | 2 |
| Screening | 13 |

Both sonography and mammogram accurately identified malignant lesions, but had false positive, 17 and 15 cases respectively. Sonography was able to correctly label three cases as benign when mammogram was concerning for malignant lesion. Also, sonography falsely labeled five masses as malignant when mammogram was suggestive of benign. Sonography helped to characterize 31 lesions as probably benign lesion or normal and obviating need for further interventions like fine needle aspiration / biopsy in 30 of the cases.

## Discussion

Retroareolar region is difficult to evaluate on mammograms as well as sonography. On mammograms, oblique nipple position and underexposure as well as difficulty in obtaining compression views may obscure lesions while variable acoustic shadows caused by the geometry and air trapped in the areolar crevices limit sonography [1, 7]. However, meticulous evaluation of this region using multiple modalities is necessary as about 8% of malignancies are seen in retroareolar region [4, 9, 10]. Tomosynthesis can be of help in evaluation of the retroareolar density, improve detection and decrease recall rate but combination of ultrasound provides more usefulness [11–14]. 3D Tomosynthesis is not as readily available as ultrasound in developing countries where mammogram availability is already limited. Having an additional ultrasound machine provides more value for money than investing for extra cost while procuring 3D tomosynthesis. ACR BIRADS categorizes asymmetry into four types, called asymmetry when seen only in one view. When seen in both views, it is called global asymmetry if involving more than a quarter of the breast; focal asymmetry if involving less than a quarter and developing asymmetry if new or larger area of asymmetry compared to a previous exam [6, 7]. The rate of malignancy increases in focal and developing asymmetries, reaching 15% [7]. Ferre R. et al, described the sonographic features of 53 retroareolar breast cancers where most (79%) were invasive ductal cancers. Of these, only 33% (17/53) were diagnosed on mammograms, mostly as pleomorphic microcalcifications, spiculated masses and distortions with fewer as focal asymmetry or mass associated with focal asymmetry [4]. None of the cases were asymmetry of types other than focal asymmetry, as described in BIRADS atlas. In our study, lesions

**Table 2. Breast density distribution in the different grades of asymmetry.**

| Grades of asymmetry | Distribution in various BIRADS density categories | | | | Total |
|---|---|---|---|---|---|
| | a | b | c | d | |
| Grade 1 | 4 | 5 | 18 | 6 | 33 |
| Grade 2 | 0 | 13 | 19 | 7 | 39 |
| Grade 3 | 4 | 9 | 14 | 1 | 28 |
| Total | 8 | 27 | 51 | 14 | N = 100 |

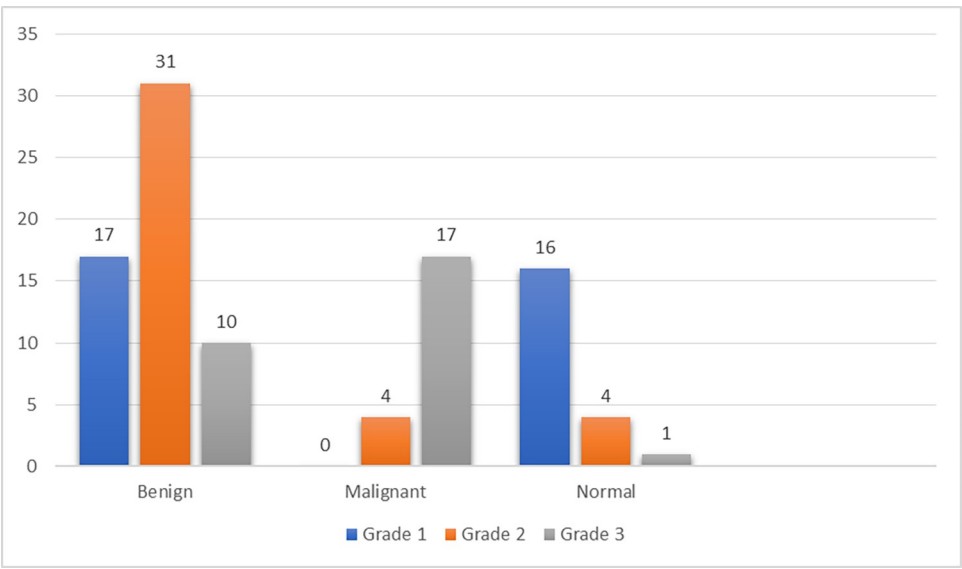

**Fig 5. Clustered bar count of grade of asymmetry and final diagnosis.**

seen as a grade 2 or 3 asymmetric density, that is, focal asymmetry with or without mass, in retroareolar area were more frequently identified as malignant on final diagnosis.

A study evaluated incidental focal asymmetry in mammograms with sonography and found mostly benign cause, with a significant relation between focal asymmetry in retroareolar region and ductal ectasia on sonography [15]. Another study evaluated the malignancy rates in retroareolar masses and intraductal abnormalities on screening breast sonography in asymptomatic women. They found the incidence of lesions in retroareolar region to be about 7.6% (87 of 1136 cases) with malignancy rate less than 2% [8]. We found a significantly greater malignancy rate (21%) in our study which may be due to our greater number of diagnostic indication among our cases and our inclusion of all the BIRADS categories unlike the prior study which included only sonographic BIRADS 3 and 4 lesions.

**Table 3. Final diagnosis of retroareolar lesions.**

| S.no | Final Diagnosis | Total Number |
|------|-----------------|--------------|
| 1. | Normal | 21 |
| 2. | Mastitis and abscess | 18 |
| 3. | Cyst (Simple or complicated) | 17 |
| 4. | Fibroadenoma | 12 |
| 5. | Ductal ectasia | 5 |
| 6. | Epidermal cyst | 1 |
| 7. | Hamartoma with fibrocystic change | 1 |
| 8. | Papilloma | 3 |
| 9. | Sclerosing adenosis | 1 |
| 10. | DCIS | 1 |
| 11. | Invasive Lobular Carcinoma | 1 |
| 12. | Invasive Ductal Carcinoma | 19 |
| | Total n = 100 Total n = 100 | 100 |

We found sonography and mammography were complementary to each other and sonography improved the certainty of benign lesions in 37 patients. Four of these patients were incorrectly labelled as suspicious malignant on mammogram. Two of these patients had grade 2 and other two had grade 3 retroareolar densities. Although no published study dedicated for retroareolar study was found in literature, studies do recommend ultrasound as useful adjunct to mammogram especially in patients with dense breasts [16–18]. In eight of our patients with suspicious mammograms, ultrasound increased the confidence of diagnosis, suggesting higher probability of cancer. These were proven to be malignant by histopathology. All except one had grade 3 asymmetric retroareolar density.

In seven of our cases, sonography overestimated the BIRADS category than suggested by mammogram, falsely increasing the probability of malignancy. Three of these had grade 3 density with final diagnosis of mastitis, papilloma and fibroadenoma, two had grade 2 with final diagnosis mastitis and two had grade 1with final diagnosis of mastitis and sclerosing adenosis. Most of our patients with asymmetric retroareolar density were symptomatic mostly with lump, nipple discharge or retraction. Among the small screening group in our study, one malignancy that is, ductal carcinoma in situ, was diagnosed seen as grade 3 asymmetric density. Due to the small size of screening group it is difficult to estimate how often asymptomatic asymmetric retroareolar density could represent malignant lesion. In our study, there was a definite trend for higher risk of malignancy with increasing grade of retroareolar density. Thus, unlike grade 1 and grade 2 asymmetry, grade 3 asymmetry increases concern for malignancy on imaging especially when associated with suspicious features like lump, nipple retraction and bloody nipple discharge. These patients need further pathological evaluation. Acquired nipple retraction especially when slit like and incomplete, may be due to benign causes like duct ectasia, periductal mastitis and tuberculosis, however, complete nipple inversion is often due to malignancy [3].

A limitation of our study is not consistently using additional mammographic views, in particular compression view. This view, though difficult in the retroareolar region due to its anatomy, may rule out parenchymal superimpositions leading to asymmetric density. It is also limited by small sample size and lack of any other similar previous studies to corroborate the findings of this study.

## Conclusion

The chances of malignancy increased with the increasing grade of asymmetric density in our study. We found grade I retroareolar asymmetric density on mammography was normal or had a benign etiology. Grade I asymmetry seen in screening mammogram may not need further evaluation. However, larger prospective studies are needed to confirm these findings. Grade 2 or 3 asymmetric need further evaluation with sonography with a pathological work up.

## Supporting information

**S1 File. Analysis retroareolar density grade and cause.**
(XLSX)

## Author Contributions

**Conceptualization:** Anamika Jha, Umesh Khanal.

**Data curation:** Anamika Jha, Ranjit Kumar Chaudhary, Shreya Shrivastav, Umesh Khanal.

**Formal analysis:** Anamika Jha, Ranjit Kumar Chaudhary.

**Investigation:** Anamika Jha, Umesh Khanal.

**Methodology:** Anamika Jha.

**Project administration:** Anamika Jha, Umesh Khanal.

**Resources:** Shreya Shrivastav.

**Software:** Ranjit Kumar Chaudhary.

**Supervision:** Umesh Khanal.

**Validation:** Shreya Shrivastav, Umesh Khanal.

**Writing – original draft:** Anamika Jha, Ranjit Kumar Chaudhary, Shreya Shrivastav.

**Writing – review & editing:** Anamika Jha, Ranjit Kumar Chaudhary, Shreya Shrivastav, Umesh Khanal.

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
