## [Decision Letter · Decision Letter 0]

13 Oct 2023

PONE-D-23-27133Ultrasonographic and Pathological Correlation of Asymmetric Retroareolar Density on MammogramPLOS ONE

Dear Dr. Chaudhary,

Thank you for submitting your manuscript to PLOS ONE. After careful consideration, we feel that it has merit but does not fully meet PLOS ONE’s publication criteria as it currently stands. Therefore, we invite you to submit a revised version of the manuscript that addresses the points raised during the review process.

We look forward to receiving your revised manuscript.

Kind regards,

Shuai Ren

Academic Editor

PLOS ONE

Journal Requirements:

2. We noted in your submission details that a portion of your manuscript may have been presented or published elsewhere. Please clarify whether this [conference proceeding or publication] was peer-reviewed and formally published. If this work was previously peer-reviewed and published, in the cover letter please provide the reason that this work does not constitute dual publication and should be included in the current manuscript.

Reviewers' comments:

Reviewer's Responses to Questions

**Comments to the Author**

1. Is the manuscript technically sound, and do the data support the conclusions?

Reviewer #1: Yes

Reviewer #2: Partly

Reviewer #3: No

2. Has the statistical analysis been performed appropriately and rigorously? 

Reviewer #1: No

Reviewer #2: N/A

Reviewer #3: No

3. Have the authors made all data underlying the findings in their manuscript fully available?

Reviewer #1: Yes

Reviewer #2: Yes

Reviewer #3: Yes

4. Is the manuscript presented in an intelligible fashion and written in standard English?

Reviewer #1: Yes

Reviewer #2: Yes

Reviewer #3: Yes

5. Review Comments to the Author

Reviewer #1: The authors aimed to identify and grade retroareolar density and correlate its grade to lesion characteristics using ultrasonography and histopathology. The manuscript is well written, and properly illustrated.

Patients and methods:

The authors described the results in frequency and percentages, without deeper statistical analysis. What was the level of significance of these frequency differences? Were there any differences/ correlations between different grades of breast density and the grade of asymmetric retroareolar density.

Conclusion: “Mild retroareolar asymmetric density on mammography is normal or has a benign etiology”. What do you mean by “mild”. It would be better to specify that grade 1 asymmetric density is normal…., as all grade 1 densities were proved to be normal or benign lesions. Please correct in the abstract and in the conclusion sections.

Reviewer #2: This is a simple review of patients presenting with central sector symptoms. Patients underwent triple assessment as per routine, and as expected the number of cancer diagnosis was in line with every symptomatic breast clinic.

The message that these low grade lesions can be discharged without further work up is quite faulty and dangerous. This study is an observational one with no follow up data on these patients. The question about how many of these patients will return with cancer, whether for missed diagnosis, or true interim primary is unknown. Symptomatic breast patients should follow triple assessment protocol as the risk of missing cancer is not acceptable. Furthermore, this study did not include other diagnostic images like tomosynthesis which might alter perspective. Clinical history and risk factor on these patients are not presented.

The conclusion of this study is simply not supported by this observational review, and is flawed and frankly dangerous. I urge the authors to review and re-consider their message carefully.

Reviewer #3: This retrospective study aimed to identify and grade retroareolar densities, correlating the grade of retroareolar density with characteristics of lesion using ultrasonography and histopathology. Retroareolar density was categorized into three grades based on morphological appearance in mammography. Sonography was performed in all patients and tissue diagnosis was obtained for suspicious lesions. After studying 100 cases, the authors concluded that mild retroareolar asymmetric density on mammography is normal or has a benign etiology and may not be evaluated further. Higher grade asymmetric density needs further evaluation with sonography with or without a pathological work up.

My comments are as follows:

a) This study has several limitations impairing its scientific importance. Retrospective analysis at a single center, with low sample size.

b) It is not clear which is the reference support to classify such asymmetries into three grades as done by the authors.

c) No information regarding expertise and number of operators performing the US, and how the lesions were assessed.

d) Important bias: inconsistency regarding the application of histopathology (only to a fraction of the sample), which may have been affected the results.

e) Such limitations do not allow us to accept the conclusions made by the authors.

6. PLOS authors have the option to publish the peer review history of their article (what does this mean?). If published, this will include your full peer review and any attached files.

Reviewer #1: **Yes: **Shaimaa Abdelsattar Mohammad

Reviewer #2: No

Reviewer #3: No

---

## [Author Response · Author response to Decision Letter 0]

29 Nov 2023

Reviewers' comments:

Reviewer's Responses to Questions

Comments to the Author

1. Is the manuscript technically sound, and do the data support the conclusions?

Reviewer #1: Yes

Reviewer #2: Partly

Reviewer #3: No

2. Has the statistical analysis been performed appropriately and rigorously?

Reviewer #1: No

Reviewer #2: N/A

Reviewer #3: No

3. Have the authors made all data underlying the findings in their manuscript fully available?

Reviewer #1: Yes

Reviewer #2: Yes

Reviewer #3: Yes

4. Is the manuscript presented in an intelligible fashion and written in standard English?

Reviewer #1: Yes

Reviewer #2: Yes

Reviewer #3: Yes

5. Review Comments to the Author

Reviewer #1: The authors aimed to identify and grade retroareolar density and correlate its grade to lesion characteristics using ultrasonography and histopathology. The manuscript is well written, and properly illustrated.

Patients and methods:

The authors described the results in frequency and percentages, without deeper statistical analysis. What was the level of significance of these frequency differences? Were there any differences/ correlations between different grades of breast density and the grade of asymmetric retroareolar density.

Answer: Due to nominal type of data, we could not perform correlation or t-test. Chi Square was performed to calculate Fisher exact test which is included in the manuscript.

Conclusion: “Mild retroareolar asymmetric density on mammography is normal or has a benign etiology”. What do you mean by “mild”. It would be better to specify that grade 1 asymmetric density is normal…., as all grade 1 densities were proved to be normal or benign lesions. Please correct in the abstract and in the conclusion sections.

Answer- Changes have been made.

Reviewer #2: This is a simple review of patients presenting with central sector symptoms. Patients underwent triple assessment as per routine, and as expected the number of cancer diagnosis was in line with every symptomatic breast clinic.

The message that these low grade lesions can be discharged without further work up is quite faulty and dangerous. This study is an observational one with no follow up data on these patients. The question about how many of these patients will return with cancer, whether for missed diagnosis, or true interim primary is unknown. Symptomatic breast patients should follow triple assessment protocol as the risk of missing cancer is not acceptable. Furthermore, this study did not include other diagnostic images like tomosynthesis which might alter perspective. Clinical history and risk factor on these patients are not presented.

The conclusion of this study is simply not supported by this observational review, and is flawed and frankly dangerous. I urge the authors to review and re-consider their message carefully.

Answer- We would like to thank reviewers for helping us provide accurate message. We meant grade I retroareolar asymmetry in screening patients may not need further sonographic assessment but agree with reviewer’s comment on limitation of this study regarding lack of follow up of these patients. So, the conclusion has been changed to reflect that. 

Reviewer #3: This retrospective study aimed to identify and grade retroareolar densities, correlating the grade of retroareolar density with characteristics of lesion using ultrasonography and histopathology. Retroareolar density was categorized into three grades based on morphological appearance in mammography. Sonography was performed in all patients and tissue diagnosis was obtained for suspicious lesions. After studying 100 cases, the authors concluded that mild retroareolar asymmetric density on mammography is normal or has a benign etiology and may not be evaluated further. Higher grade asymmetric density needs further evaluation with sonography with or without a pathological work up.

My comments are as follows:

a) This study has several limitations impairing its scientific importance. Retrospective analysis at a single center, with low sample size.

b) It is not clear which is the reference support to classify such asymmetries into three grades as done by the authors.

c) No information regarding expertise and number of operators performing the US, and how the lesions were assessed.

d) Important bias: inconsistency regarding the application of histopathology (only to a fraction of the sample), which may have been affected the results.

e) Such limitations do not allow us to accept the conclusions made by the authors.

Answer: We agree with the reviewer’s comment about the limitations of the study and has been elaborated to help reviewer understand the limitation of the study. The conclusion has been modified to reflect that. We agree with potential bias from not all patients undergoing biopsy but would be unethical for patients with benign or probably benign diagnosis to refer for biopsy. The conclusion of the study has been modified to reflect such limitations.

The information on US operator has been updated and information on mammogram interpreters was included in material and methods section. 

The grade system has been used for convenience and not used in BIRADS lexicon. The definitions of the grades are based on BIRADS lexicon and included in the manuscript.

6. PLOS authors have the option to publish the peer review history of their article (what does this mean?). If published, this will include your full peer review and any attached files.

Do you want your identity to be public for this peer review? For information about this choice, including consent withdrawal, please see our Privacy Policy.

Reviewer #1: Yes: Shaimaa Abdelsattar Mohammad

Reviewer #2: No

Reviewer #3: No

---

## [Decision Letter · Decision Letter 1]

8 Jan 2024

PONE-D-23-27133R1Ultrasonographic and Pathological Correlation of Asymmetric Retroareolar Density on MammogramPLOS ONE

Dear Dr. Chaudhary,

Thank you for submitting your manuscript to PLOS ONE. After careful consideration, we feel that it has merit but does not fully meet PLOS ONE’s publication criteria as it currently stands. Therefore, we invite you to submit a revised version of the manuscript that addresses the points raised during the review process.

We look forward to receiving your revised manuscript.

Kind regards,

Shuai Ren

Academic Editor

PLOS ONE

Reviewers' comments:

Reviewer's Responses to Questions

**Comments to the Author**

1. If the authors have adequately addressed your comments raised in a previous round of review and you feel that this manuscript is now acceptable for publication, you may indicate that here to bypass the “Comments to the Author” section, enter your conflict of interest statement in the “Confidential to Editor” section, and submit your "Accept" recommendation.

Reviewer #1: All comments have been addressed

Reviewer #2: All comments have been addressed

2. Is the manuscript technically sound, and do the data support the conclusions?

Reviewer #1: Yes

Reviewer #2: Yes

3. Has the statistical analysis been performed appropriately and rigorously? 

Reviewer #1: Yes

Reviewer #2: N/A

4. Have the authors made all data underlying the findings in their manuscript fully available?

Reviewer #1: Yes

Reviewer #2: Yes

5. Is the manuscript presented in an intelligible fashion and written in standard English?

Reviewer #1: Yes

Reviewer #2: Yes

6. Review Comments to the Author

Reviewer #1: in page 7: "Using Fisher’s exact test, the odds

126 of grade 1 asymmetry to be a malignant lesion was significantly low (p= 0.00)". It would be appropriate to write the p value as <0.001

Reviewer #2: Thank you for addressing the comments. I think including the raw table of statistical analysis is not appropriate for a publication. The final p-value should be put in the data table and the manuscript should just state which test was used without including the raw statistical program tabulation of the analysis. The software, and its version, used for statistical analysis should also be stated in the methodology

I think more emphasis should be given on the 21 patients who were diagnosed with malignancy, especially the four patients who the author described as having grade 2 lesions on radiology. Their presentation, work up and surgery and further treatment should be elaborated on to add value to the paper.

7. PLOS authors have the option to publish the peer review history of their article (what does this mean?). If published, this will include your full peer review and any attached files.

Reviewer #1: No

Reviewer #2: No

---

## [Author Response · Author response to Decision Letter 1]

13 Feb 2024

Reviewers' comments:

Reviewer's Responses to Questions

Comments to the Author

1. If the authors have adequately addressed your comments raised in a previous round of review and you feel that this manuscript is now acceptable for publication, you may indicate that here to bypass the “Comments to the Author” section, enter your conflict of interest statement in the “Confidential to Editor” section, and submit your "Accept" recommendation.

Reviewer #1: All comments have been addressed

Reviewer #2: All comments have been addressed

2. Is the manuscript technically sound, and do the data support the conclusions?

Reviewer #1: Yes

Reviewer #2: Yes

3. Has the statistical analysis been performed appropriately and rigorously?

Reviewer #1: Yes

Reviewer #2: N/A

4. Have the authors made all data underlying the findings in their manuscript fully available?

Reviewer #1: Yes

Reviewer #2: Yes

5. Is the manuscript presented in an intelligible fashion and written in standard English?

Reviewer #1: Yes

Reviewer #2: Yes

6. Review Comments to the Author

Reviewer #1: in page 7: "Using Fisher’s exact test, the odds

126 of grade 1 asymmetry to be a malignant lesion was significantly low (p= 0.00)". It would be appropriate to write the p value as <0.001

Author’s Response: Thank you for the feedback. Change has been made to address the comment (line 105). 

Reviewer #2: Thank you for addressing the comments. I think including the raw table of statistical analysis is not appropriate for a publication. The final p-value should be put in the data table and the manuscript should just state which test was used without including the raw statistical program tabulation of the analysis. The software, and its version, used for statistical analysis should also be stated in the methodology

I think more emphasis should be given on the 21 patients who were diagnosed with malignancy, especially the four patients who the author described as having grade 2 lesions on radiology. Their presentation, work up and surgery and further treatment should be elaborated on to add value to the paper.

Authors’ response: The table has been removed and data has been presented in the text (line 105). Additional information on clinical course of the patients with malignancy have been included (Line 107-114) 

7. PLOS authors have the option to publish the peer review history of their article (what does this mean?). If published, this will include your full peer review and any attached files.

Do you want your identity to be public for this peer review? For information about this choice, including consent withdrawal, please see our Privacy Policy.

Reviewer #1: No

Reviewer #2: No

---

## [Decision Letter · Decision Letter 2]

13 Mar 2024

Ultrasonographic and Pathological Correlation of Asymmetric Retroareolar Density on Mammogram

PONE-D-23-27133R2

Dear Dr. Chaudhary,

We’re pleased to inform you that your manuscript has been judged scientifically suitable for publication and will be formally accepted for publication once it meets all outstanding technical requirements.

Kind regards,

Shuai Ren

Academic Editor

PLOS ONE

Additional Editor Comments (optional):

Reviewers' comments:

Reviewer's Responses to Questions

**Comments to the Author**

1. If the authors have adequately addressed your comments raised in a previous round of review and you feel that this manuscript is now acceptable for publication, you may indicate that here to bypass the “Comments to the Author” section, enter your conflict of interest statement in the “Confidential to Editor” section, and submit your "Accept" recommendation.

Reviewer #1: All comments have been addressed

2. Is the manuscript technically sound, and do the data support the conclusions?

Reviewer #1: Yes

3. Has the statistical analysis been performed appropriately and rigorously? 

Reviewer #1: Yes

4. Have the authors made all data underlying the findings in their manuscript fully available?

Reviewer #1: Yes

5. Is the manuscript presented in an intelligible fashion and written in standard English?

Reviewer #1: Yes

6. Review Comments to the Author

Reviewer #1: review and you feel that this manuscript is now acceptable for publication. The authors addressed what was needed regarding the p value , and statistical analysis.

The manyscript was written in clear and correct language.

I have no further comments.

7. PLOS authors have the option to publish the peer review history of their article (what does this mean?). If published, this will include your full peer review and any attached files.

Reviewer #1: No

---

## [Author Response · Author response to Decision Letter 2]

5 May 2024

Reviewers' comments:

Reviewer's Responses to Questions

Comments to the Author

1. If the authors have adequately addressed your comments raised in a previous round of review and you feel that this manuscript is now acceptable for publication, you may indicate that here to bypass the “Comments to the Author” section, enter your conflict of interest statement in the “Confidential to Editor” section, and submit your "Accept" recommendation.

Reviewer #1: All comments have been addressed

Reviewer #2: All comments have been addressed

2. Is the manuscript technically sound, and do the data support the conclusions?

Reviewer #1: Yes

Reviewer #2: Yes

3. Has the statistical analysis been performed appropriately and rigorously?

Reviewer #1: Yes

Reviewer #2: N/A

4. Have the authors made all data underlying the findings in their manuscript fully available?

Reviewer #1: Yes

Reviewer #2: Yes

5. Is the manuscript presented in an intelligible fashion and written in standard English?

Reviewer #1: Yes

Reviewer #2: Yes

6. Review Comments to the Author

Reviewer #1: in page 7: "Using Fisher’s exact test, the odds

126 of grade 1 asymmetry to be a malignant lesion was significantly low (p= 0.00)". It would be appropriate to write the p value as <0.001

Author’s Response: Thank you for the feedback. Change has been made to address the comment (line 119). 

Reviewer #2: Thank you for addressing the comments. I think including the raw table of statistical analysis is not appropriate for a publication. The final p-value should be put in the data table and the manuscript should just state which test was used without including the raw statistical program tabulation of the analysis. The software, and its version, used for statistical analysis should also be stated in the methodology

I think more emphasis should be given on the 21 patients who were diagnosed with malignancy, especially the four patients who the author described as having grade 2 lesions on radiology. Their presentation, work up and surgery and further treatment should be elaborated on to add value to the paper.

Authors’ response: The table has been removed and data has been presented in the text (line 119). The test performed and software used has been added (line 94) Additional information on clinical course of the patients with malignancy have been included (Line 121-128) 

7. PLOS authors have the option to publish the peer review history of their article (what does this mean?). If published, this will include your full peer review and any attached files.

Do you want your identity to be public for this peer review? For information about this choice, including consent withdrawal, please see our Privacy Policy.

Reviewer #1: No

Reviewer #2: No

---

## [Editor Report · Decision Letter 3]

10 Oct 2024

Ultrasonographic and Pathological Correlation of Asymmetric Retroareolar Density on Mammogram

PONE-D-23-27133R3

Dear Dr. Chaudhary,

We’re pleased to inform you that your manuscript has been judged scientifically suitable for publication and will be formally accepted for publication once it meets all outstanding technical requirements.

Kind regards,

Shuai Ren

Academic Editor

PLOS ONE
---

## [Editor Report · Acceptance letter]

26 Oct 2024

PONE-D-23-27133R3 

PLOS ONE

Dear Dr. Chaudhary, 

I'm pleased to inform you that your manuscript has been deemed suitable for publication in PLOS ONE. Congratulations! Your manuscript is now being handed over to our production team.

Kind regards, 

on behalf of

Dr. Shuai Ren 

Academic Editor

PLOS ONE